# English Development Sustainability for English as Second Language College Transfer Students: A Case Study from a University in Hong Kong

Dennis Foung [1], Shirley Siu Yin Ching [2], Lillian Weiwei Zhang [2], Gwendoline Yuanyuan Guan [2] and Kin Cheung [2,*]

1   School of Journalism, Writing, and Media, The University of British Columbia, Vancouver, BC V6T 1Z4, Canada
2   School of Nursing, The Hong Kong Polytechnic University, Hong Kong, China
*   Correspondence: kin.cheung@polyu.edu.hk

**Abstract:** The sustainability of English development plays a crucial role in higher education. However, the language needs of community college transfer students have not been well studied. This paper examined the language needs and support measures for vertical transfer (VT) English as a Second Language (ESL) students after admission to the university. A qualitative approach was adopted. Thirty-nine focus groups and seven individual interviews were conducted with 124 VT ESL students. The results found that, while community college studies might have prepared VT students for basic written assignments in universities, these students needed support with advanced academic writing skills, and general speaking and listening skills. It is only if the needs and challenges of VT ESL students are clear to higher education administrators that effective strategies can be developed. For instance, the participants were not content with the current measures provided to them and required short, fun, and purpose-driven interventions. This is the first of its kind to explore the English needs and support measures among VT ESL to sustain their English development should be strengthened.

**Keywords:** vertical college transfer students; university students; non-native English speakers; language needs and support; qualitative study; English sustainability

## 1. Introduction

Transferring from community college to university provides an accessible and indirect route for educational expansion and brings about changes to the landscape of higher education. Vertical transfer (VT) students are those who have obtained post-secondary qualifications and have been admitted to pursue bachelor's degrees in universities [1]. Through the recognition of prior learning in community colleges, VT students are usually required to spend another two to three years completing their bachelor's degrees [2]. With the promotion of internationalization in universities around the globe, worldwide, universities increase the admission of transfer students [3]. Some of these students speak English as a second language (ESL), having acquired some qualification (and perhaps an English pathway program) before being admitted to English-speaking universities. While the needs for English development from community college to university are crucial, the needs of VT ESL students in English support have not been well studied.

### 1.1. Literature Review

Past research has already established that transfer students may encounter a number of barriers, especially in English sustainability. For example, Robison, Fawley and Marshall [4] found that some VT ESL students did know that they needed to take extra English courses, but they received conflicting information that left them confused about what they actually needed to do. How to sustain VT ESL students' English development is an issue.

Frodesen [5], while running a course for VT ESL students, found that they might have taken courses in academic English but that most of the time was spent working on source-based writing. Many other important aspects could not be covered due to time constraints. Lin and Yi [6] warned that these ESL students with problems in English can end up with trouble when it comes to completing their assignments. These assignment issues can eventually become major causes of psychological stress in their university studies, together with many other stressors caused by college transfer. Even though English is known to be an important and possibly problematic factor for ESL students during the college transfer process, past research has only managed to confirm English as a factor, but there has been limited detail revealed about the needs of these VT ESL students.

Other than the language ability of these transfer students, their confidence and motivation seem to be an issue as well. Castro and Cortez [7] found that Mexican ESL transfer students, in general, did not think that they had the language proficiency to complete their university programs. The authors believed that these students were just lacking in confidence, as they did not have any way to prove their proficiency levels. Furthermore, while there is not much evidence to describe transfer students' motivation to improve their English skills, there is a wide range of evidence to support that they need to deal with many transfer problems immediately after their university admission. For example, transfer students encounter campus culture shock [8], and they need to adjust to university study and establish new relationships with peers and faculty members. Perhaps, as one of all these tasks, learning English has not been a motivating task [9].

Unlike VT ESL students, the language needs of ESL students who are admitted to the university directly from their secondary schools have been researched thoroughly. These students have been referred to in the literature as "direct entrants" [8]. Evans and Morrison [10], for example, conducted an extensive study to understand the language needs of ESL direct entrants in Hong Kong. They asked students to rank the levels of difficulty of each of the four skills in English sustainability, and writing was ranked as the most difficult, followed by reading, speaking and listening. Another study by Evans and Morrison [11] revealed that speaking accurately (grammar) and understanding specialist vocabulary (reading) were ranked highly amongst sub-skills (i.e., across all four skills) as difficult micro-skills. Three writing sub-skills were ranked as the most difficult, including using appropriate style, linking sentences smoothly, and using grammar correctly. Further interviews in their study indicated that students had problems with their professors' requirements and expectations, written assignments, dealing with disciplinary genres, and planning long written assignments. Their results were similar to those of Lobo and Gurney [12], with ESL university students in Australia. The students in their study believed that writing skill development is necessary, followed by speaking skills, and learning new technical vocabulary. They also saw the need to improve communication skills, grammar, and referencing in sustaining their English ability. Most English for academic purposes (EAP) courses are subsequently developed based on the needs of this group of mainstream students.

Direct entrants' motivation to learn the English language has been researched well. Snow and colleagues [13] claimed that "self-theories" affect this motivation. Some students believed that their ability (i.e., English proficiency) could not be controlled, hence they might be less motivated to improve their English proficiency. Those who believed that they could improve their language skills would be motivated to do so. Other studies of students' behaviors during the first year of EAP courses found that they are "pragmatic" in doing EAP coursework [14]. Some could even adopt goal-oriented approaches to complete their assignments, i.e., only meeting the goals and paying no/little attention to the learning process [15].

### 1.2. Research Questions

While past research seems to be able to reveal that there were some similarities and differences in the language needs of ESL direct entrants and VT students, as outlined above,

there seems to be a need to extend the understanding of the needs of VT ESL students. Both VT ESL students and direct entrants may be taking the same courses, with the same intended learning outcomes, assessments and learning activities, and these may not address the needs of VT ESL students to sustain their English development. More specific research for this group is addressing the following questions:

1.  What are the language needs of VT ESL students?
2.  What can university language centers do to address the needs of this group of students?

The rationale for these questions is the following. First, it is only if the English needs of VT ESL students are clear to higher education administrators that sufficient and specific support can be provided. Second, university English teachers need to know how they can adjust their teaching to fit their specific needs.

## 2. Materials and Methods

### 2.1. Study Design

A focus group approach was chosen as the method of data collection because it allows participants to discuss complex subjects in amenable settings and facilitates them to reveal what they think and why they think what they do [16]. Graneheim and Lundman [17] described a procedure for conducting qualitative content analysis to obtain both a broad and an in-depth understanding of the phenomenon in question [18]. Institutional ethical clearance has been obtained from the study university (HSEARS20170808003).

### 2.2. Participants and Setting

Purposive sampling was adopted to recruit participants from a university that admitted the highest number of VT ESL students in Hong Kong. The selection criteria included: (a) VT students who had graduated from a two-year community college and were studying in or had graduated from a bachelor's program of various disciplines; (b) those who had been studying in the program for at least one semester; (c) students of Chinese ethnicity; and (d) those who spoke English as a second language (i.e., used Chinese in their daily lives, but English in a wide range of contexts in daily life). A total of 2,143 current college transfer students and 20 graduates who graduated one or two years previously from the two/three-year bachelor's program were invited through emails and their program leaders. Among them, 130 participants (6.0 %) who replied to the emails were invited to join focus groups in a quiet room at the research site.

### 2.3. Data Collection

A focus group is a method of collecting data on a designated topic by using a semi-structured group session that is moderated by an interviewer in an informal setting [19]. Thirty-nine focus groups and seven individual interviews were conducted with 115 current and nine graduated VT ESL students from February 2018 to March 2019. In total, 40 males and 84 females, involving students from multiple disciplines (including Applied Sciences, Building Sciences, Business, Engineering, Health Sciences, Hotel Management, and Humanities) and all years of study, joined the interviews. The group size ranged from two to five. Most groups consisted of 3–4 participants depending on the student's availability. The interviews lasted for 30–45 min. Each student was interviewed once by an interviewer with experience in qualitative studies, using an interview guide (Table 1). One research assistant joined each focus group and noted the interactions among the participants and both of them were not affiliated with the interviewees (i.e., not being a teaching staff member in any courses).

**Table 1.** Interview guide for exploring the language needs and support measures for ESL VT students.

| General Broad Opening Question | |
|---|---|
| 1. | Can you tell me about your experience on learning English, till now, in the university? |
| Probing questions | |
| 1. | Have you experienced any problems in learning or use of English? |
| 2. | Have you experienced any problems in studying your discipline that is related to English? |
| 3. | In terms of the use of English, which area do you think you need help with? |

All interviews were conducted in Cantonese (the students' first language). A funnel-based approach was adopted [20] during the interview. The interviews started with "We have some questions about English use. First of all, in general, please describe your English learning experience in the university so far?"

Probing questions were asked, which allowed the participants to elaborate on their experiences with English use. The interviewer was active in creating group discussions for the data collection [21].

*2.4. Data Analysis*

Manifest qualitative content analysis [18] was used in analyzing the interview data. The audio recordings of the interviews were transcribed verbatim in Chinese and then imported into NVivo Pro 12 for data management and analysis. According to Graneheim and Lundman [17], qualitative content analysis is useful to emphasize the context and subject which are essential for the focus group interview transcripts in the current study. A typical content analysis consisted of five steps: read and re-read the interview transcriptions; identify the meaning units (sentences/paragraphs) corresponding to different aspects of the student's experiences as they emerged in the data; condense meaning units and label them with code; identify subcategories by comparing their similarities and differences, and delineate the key categories [17]. The interviewer and the research assistant analyzed the data independently and then checked the coding for consistency until a consensus was reached. The trustworthiness of the study was established by allowing a qualified researcher to review the transcripts and coding through peer debriefings and recording the steps and decisions made during data analysis through an audit trail [22]. Details of the data collection, analyses and strategies to ensure the trustworthiness of the findings have been described in another manuscript [23]. Selected quotations from the interview were translated into English for reporting purposes by a bilingual translator.

**3. Results**

The results of all individual and focus group interviews were grouped under six categories: (1) Writing and Reading Needs; (2) Speaking and Listening Needs; (3) Grammar and Vocabulary Needs; (4) Perceived Difference between VT and Direct Entrants; (5) Motivation for Learning English; and (6) Perceptions of Support Measures.

*3.1. Students' Writing and Reading Needs*

Generally speaking, the students were not facing major challenges in reading and writing. Even though a few students mentioned the need for top-down reading skills, these skills did not emerge as a theme in the data. The students had learnt many of the necessary writing skills (e.g., paragraph development, style, and structure) in their college English courses.

> *[We learnt] some language [skills] needed for paper writing. And the papers we usually write are academic in style; this type of paper tends to avoid personal language such as 'I' or 'we'. (46, A2, Hotel & Tourism, Year2)*

*[We] probably [learnt] organizational structure, thesis statement, what to include in the introduction, body, and conclusion. Also, [learning] how to write topic sentences and how to elaborate, are all very useful. (16, D, Humanities, Year2)*

However, the students seemed to have problems with their Final Year Projects, which are most often expected to be research papers.

*For example, the thesis [Final Year Project] has a discussion section. We have to develop the ideas ourselves, but sometimes it's difficult to think about ideas, and we don't know if our ideas are right or not. (03, A, Hotel & Tourism, Year2)*

*And we don't know which style is really right, because every paper seems to be different in style, but is there a relatively better one? I think teachers can give us some suggestions so that we would know what is preferred and we can work towards it accordingly. (07, A, Health Sciences, Year2)*

*The tone, because I've never done a research paper before. I am unsure of the tone of language, so I have to refer to others' English writing, but every paper is different, and I don't really know if there are any guidelines to follow. (19, B, Health Sciences, Year3)*

### 3.2. Speaking and Listening Skills

The students were more concerned with their listening and speaking skills, but their worries were related more to the communication aspects in a general setting, not the use of these skills in an academic context.

In terms of speaking, the interesting trend was that many courses (and even disciplines) did not require a speaking assessment. Where spoken assessments did take place, they were most likely to be presentations in which they could adopt strategies they were used to. Other than these, the students did not find other opportunities to interact with others in English.

*I think speaking [is challenging], because in the past 4 to 5 years since my sub-degree study, we have learnt writing and reading, . . . we are used to them. But in terms of speaking there are really few opportunities. We can only practice speaking for assessment, even only for the assessment in an English subject. There are no such opportunities for other subjects which mostly involve only written assessment. We have difficulty in listening and understanding accents. It is very important to understand different accents in the future when speaking to others. However, we are currently required to perform writing or speaking assessment only. (35, S2, Business, Year1)*

*I think speaking is the most important of all the English skills. I want to improve it constantly, but there are few opportunities to practice speaking here at the university, because few subjects in the Accounting Departments require a [oral] presentation. Even after we present [there is barely feedback on our speaking/presentation], if we keep using the same way to present, we will use it all along, and there won't be any improvement . . . I think there are few opportunities in class for us to practice speaking. (15, A, Business, Year2)*

*. . . Actually I need to improve in all the areas. For example, we rarely interact with others in English. Most of the time we just do presentations, but this is one-way communication and relatively formal, and all related to business. We rarely use English to talk to people. (25, S1, Business, Year2)*

When students saw the need for speaking and listening, e.g., during an internship, they are most worried about their accents and their ability to communicate an idea. They seem to lack confidence.

*I think speaking is the most difficult, because sometimes you don't know how to express your ideas. There are a couple of possible outcomes [in communication], and probably the most terrible one is that you want to say something, but you can't use the right words to express it exactly. Another possible outcome is that I am afraid to speak English, because I*

*will panic while speaking. I think I can't talk fluently and have an accent. (33, S2, Social Sciences, Year1)*

In terms of listening skills, the students once again expressed the belief that they are competent in an academic context (i.e., attending lectures) because they are used to it. However, when it comes to everyday communication, students do not understand when the speakers express themselves colloquially (e.g., slang and phrasal verbs). This, once again, happens in a non-academic context, such as communicating with clients during internships or watching movies.

*My listening is poor. We have to meet many international customers during the internship. Probably instead of having a problem with their accents, I have problems with the vocabulary they use, that is, they use different words to refer to the same meanings, but I may not know these words. (46, A2, Hotel & Tourism, Year2)*

*Perhaps I can't get the slang, or phrasal verbs they use. (46, A1, Hotel & Tourism, Year2)*

*I'm relatively weak in listening. I have no major problem listening to lectures, because we are all Hong Kongers and I am used to the English we use, but when it comes to English drama, I can't understand what they say when I only watch it without subtitles. I feel my English is just so so. The accents and the paces of speech that foreigners have in their countries are more complex. This problem becomes quite apparent when I watch dramas. (01, A5, Health Sciences, Year2)*

### 3.3. Grammar and Vocabulary Needs

These two categories (Grammar and Vocabulary) belong to the language system categories, i.e., general language proficiency. The students did not seem to have problems with grammar (i.e., not their needs) and no major grammar-related themes emerged. However, they were not familiar with the specialist vocabulary in either listening or reading contexts. This means that such vocabulary can be a major barrier to their comprehension when attending lectures or reading discipline-based articles.

*I think I do not know sufficient technical terms or vocabulary of our discipline. Sometimes I understand what the teachers say but I can't spell the terms. I think it's a big problem, because I can't spell the words according to their pronunciation. (07, A, Health Sciences, Year2)*

*Some students might have attended secondary schools with Chinese as the medium of instruction; they will have difficulty with classes in English at university. They have to check the dictionary when reading PowerPoint slides, while others can understand them instantly. (20, B, Applied Sciences, Year2)*

### 3.4. Perceived Difference between VT and Direct Entrants Students

Interestingly enough, when the VT ESL students were asked to compare themselves with direct entrants, only grammar and vocabulary issues were mentioned. In other words, the perceived differences between these two groups seem to be with the general language issues.

*When I did a group project with the first-year-first-degree [FYFD] students, they could instantly type out our discussion and post it to BlackBoard [the university learning management system]. I felt shocked as they could type up one paragraph so quickly without checking the grammar. (01, A2, Health Sciences, Year2)*

*Their [FYFD students] writing, grammar and vocabulary skills are better than ours. When you read their writing, maybe there are problems in content or concept, but there are a few grammar mistakes. (25, S2, Business, Year2)*

### 3.5. Motivation for Learning English

The language learning motivation of students was diverse. Some students were motivated to learn English because they saw a very specific need for them to improve (e.g., reading English notes, writing English papers, and writing complex sentences).

*Now everything is in English, so there is a greater need to learn English. In contrast, not all the subjects in the associate degree were in English; some notes were in Chinese. (33, S3, Social Sciences, Year1)*

*My English is not good, and the papers we have to write at university are more demanding. I have the motivation to improve. (04, A2, Health Sciences, Year2)*

While some students see the need to improve their English skills, some other students described English as a small part of university study and they had many other courses to work on (for better grades). That is why improving English is not a major task.

*Because we don't have confidence in the first place, and we have to care about the overall GPA. How can we improve English under these circumstances? You have to give up something if you want to improve English, probably the subject grade, so I choose not to spend time on English. (05, A3, Health Sciences, Year1)*

*But I think we have limited time to work on so many things, studying and part-time jobs, we don't have time to improve our English. (15, B, Business, Year2)*

*um . . . just no time, because even if we have time we would work on major subjects and assignments rather than English. (18, A1, Business, Year2)*

### 3.6. Perceptions of Support Measures

The VT ESL students' perceptions of the support offered by the university were not positive. One of the possible reasons is that they were busy and had packed schedules. If the support offered does not fit into their packed schedules, they are not able to obtain it.

*Actually, there is support provided by the university, but will you go for it? We're busy and will not seek help from them. I heard about their briefing, but didn't find anything interesting to me. (04, A2, Health Sciences, Year2)*

*We receive BlackBoard notices or emails about English courses from the university. But normally we don't attend them due to time constraints or packed schedules. (21, A, Health Sciences, Year3)*

Not surprisingly, the services that were long in duration, or not tailor-made to students' individual problems, were criticized.

*The DELTA [test] is lengthy, I really wanted to answer the questions randomly towards the end of the test. (07, C, Health Sciences, Year2)*

*I had wanted the mentor to tell me my specific problem. I had thought it would be difficult to improve overall, but I didn't know which area needed more work and which didn't. So I joined the [Excel in English] Scheme with the intention that the mentor would know which area to focus on for me. However, it became a group discussion like those we did in secondary school. Probably it was just this mentor, other mentors might be different. (01, A2, Health Sciences, Year2)*

Instead of running academic-based support services, the students suggested having more non-academic activities.

*Organise some activities, like the workshops on recognizing wrong words and pronunciation conducted by the Department of Chinese and Bilingual Studies. I attended their sessions, and they gave us coupons as rewards. These activities are attractive. They also have the Open Day of the Confucius Institute on Saturday. I think probably the English Language Centre could also do Shakespeare. (38, S3, Business, Year1)*

*Can they organize some fun activities, rather than those academic ones, such as improving IELTS? Anyway, the titles of activities always sound boring and are related to improving English or some skills. I think we won't attend, because sitting in the workshop for 3 h to improve skills sounds like seminars or lectures, and you won't feel it's fun. It is not as interesting as recognizing wrong words. (38, S2, Business, Year1)*

*Their activities are really for those who have a huge motivation to improve, like the Big Mouth Corner, or the one on how to improve writing. Seems like there are some pen-friend activities. Only really highly motivated students will participate in these activities. I think the game-type of activities with more amusement would attract more people. (32, S2, Building Sciences, Year2)*

To summarize the needs of the VT ESL students, first, they need more support with advanced skills in academic writing to deal with what they need to handle in their senior-year studies. Second, they also have more practical needs in listening and speaking skills with regard to English communication (i.e., instead of needs for academic purposes). Third, they also need help with discipline-related vocabulary. Unfortunately, they were not highly motivated to work on their English skills due to their busy schedules and the support provided by the university has not been appealing to some of them.

## 4. Discussion and Implications

This is the first of its kind of study to explore the English needs and support measures among VT ESL students. It is only if the needs and challenges of VT ESL students are clear and specific to higher education administrators that useful and effective support measures can be provided to this group of students to sustain their English development, particularly on seamless articulation from community college to university.

### 4.1. RQ 1: Language Needs of VT ESL Students

#### 4.1.1. Language Learning Motivation

The findings of our study revealed that the VT ESL students' motivation to learn English at university was not high. This might be attributed to their previous experiences with English. Prior studies have shown that negative experiences with learning English in secondary school can have a long-lasting impact on students' attitudes towards language sustainability [24]. The VT ESL students were mostly taught in an exam-oriented culture [25], whereby monotonous and repetitive exercise drilling and test preparation occupied a great majority of English learning, and this, unfortunately, has undermined their interest in the language and its culture. One plausible explanation is that when English proficiency is not counted towards their final GPA in university, students would not put effort into it. Another possible explanation for the low motivation of VT ESL students can be attributed to their capability to be strategic in their university studies. They have acquired "time management" skills in their college studies, i.e., to prioritize subjects and assignments according to their importance [26]. That is how they learn to be strategic. As English is only one of the subjects and they have other "transfer shock" and "campus shock" issues to deal with [9,27,28], along with their busy schedules (as indicated in the results), improving English becomes a lower priority item. This echoes the findings of a previous study with a comparable group of students [23].

#### 4.1.2. Needs for Speaking and Listening Skills for Non-Academic Context

Many of the interviewees thought they had sufficient listening and speaking skills to deal with their academic needs, but they did see the need to improve these skills (e.g., accent and pronunciation) in non-academic contexts. The obvious reason for this was that there are limited opportunities to speak English in a Cantonese-dominated society, especially when they lack the confidence to communicate clearly and fluently [9]. Although they are required to use English only in class, out-of-class communication including project and study-group communication is generally conducted in Cantonese. Having said that, the students did describe having sufficient opportunities to attend lectures presented in

English and to complete the oral assessment in English during their college studies, so that they felt confident about using it sufficiently to meet their academic needs.

The students could see the need for listening and speaking in non-academic contexts. In fact, many of the interviewees supported this viewpoint by citing examples from their internships. This need for non-academic English was not considered by Evans and Morrison [11] in their studies of direct entrants; the students in our study needed to plan for (or had already completed) internships as VT ESL students and some of them might also want to go abroad for exchange programs. These are common experiences in university for students in their final year of study, which explains why our interviewees described listening and speaking needs in non-academic contexts. The positive implication for this would be that the internship/exchange program indeed motivates them to improve their English. If there is no such program, these students may be content with their current levels of listening and speaking ability.

### 4.1.3. Needs for Writing Skills for Advanced Studies

The interviewees in our study indicated that they had acquired essential writing skills for their general writing assignments, but that they needed more help with their final year projects. The interviewees explicitly mentioned the usefulness of the "structure", "format" and "academic tone" learnt in their college studies. Evans and Morrison [11] reported the needs of direct entrants as being: "unfamiliarity with disciplinary genres and referencing conventions", "inexperience in planning and writing extended texts requiring the synthesis of information and ideas from multiple sources", and "apparent inability to communicate their understanding of the subject matter in stylistically appropriate academic prose". The experiences of our interviewees suggest that they have already addressed the needs experienced by direct entrants identified in other studies. However, as revealed by Pullen [29], VT ESL students were exposed during their college years to one or two courses to prepare them for generic source-based writing. This, however, only prepared them for a "transfer-level research course" [29] (p.20). This may explain why VT ESL students need help with research courses, such as the final year project. This has clear implications regarding the type of support to be offered.

### 4.1.4. Need for Specialist Vocabulary

The participants in our study expressed the need to know more specialist vocabulary for comprehension purposes. Interestingly enough, this aligns with many other studies about the needs of direct entrants [11,30,31]. Quite a number of other studies reported specialist vocabulary to be a key need of university students in general [32–34]. In reality, direct entry nursing students start having problems with specialist vocabulary in year one courses but VT ESL students have faced and acquired these words in their college studies. In the senior year of direct entrants and VT ESL students, they may encounter some other new specialist vocabulary in senior year courses. Other disciplines also have other specialized courses in both freshmen year and senior year. As long as students are learning something new and specialized in their courses, all students need to learn some specialist vocabulary.

As suggested above, learning new specialist vocabulary seems to be a never-ending problem, it seems that one possible way to solve this would be to ask students to acquire strategies to work with specialist vocabulary. The current research seems to look for ways to help learners acquire vocabulary effectively through explicit instruction, such as classroom activities with pre-task, task and assessments (See Gablasova [35] as an example). However, some other more innovative ways may be more useful to help these students, given that they are busy and have low motivation in language learning.

### 4.1.5. Perceived Differences between Direct ESL Entrants and VT ESL Students and Their Impacts

This study revealed the perceived differences between the two groups of students. While these are only perceptions, and hence may not reveal the actual differences (which require an objective assessment), they deserve further discussion. As reported in the results, the students perceived a difference in general language proficiency. The VT ESL students perceived the direct entrants to be better in writing, skills and grammar accuracy, and to have larger vocabulary banks. However, the examples quoted by the VT ESL students were often authentic/spontaneous situations (e.g., in-class discussions). The obvious reason is that VT ESL students can only compare by observing others in the class and it may not be possible for them to actually read the others' assignments. While the actual difference remains a question, the VT ESL students' observation seems to have an impact on their confidence in the general use of English.

Recognising the problems, or perhaps their lack of confidence, with their general English communication skills, VT ESL students are more forward-looking and think about their language needs for their internship and exchange programs. With such a forward-looking attitude, VT ESL students start improving themselves much earlier than direct entrants. However, these specific reasons for motivation and forward-looking attitudes have not been reported in previous studies of direct entrants [10,11].

### 4.2. RQ2: Support Measures for VT ESL Students

The interviews in the current study revealed that VT ESL students prefer support measures that are non-academic, quick, purpose-driven, and fun. These measures also need to fit into their packed schedules. The needs of VT ESL students in different language areas need to be addressed as well.

First, to address the language needs for Final Year Report writing and perhaps the specialist vocabulary needs, the university can offer a range of Writing Across the Curriculum support measures, not only to VT ESL students, but also to direct entrants. This could be in the form of mobile applications, for example, learning discipline-specific vocabulary [36], supporting the writing of the Final Year Project [37], or it could be a series of workshops on the final year project. Short-term support measures could include some tangible targets for participants to achieve in a fun way (e.g., 10 new words for a specific discipline). Longer initiatives should provide sufficient motivation for students to continue to use them (e.g., specific tips on improving the grades for the Final Year Report). These can meet the student's academic needs while giving full consideration to their logistical constraints and motivation.

Other than these academic provisions, non-academic sessions should be organized to boost the student's confidence and speaking/listening skills. While Robison et al. [4] reported that there was a lack of proper orientation support for VT ESL students, the university (or the English Language center) could organize an English-speaking buddy program for VT ESL students. Each meeting could be around a specific aspect of VT transfer shock, such as course selection, applying for exchange programs, completing internship requirements or credit transfer applications. These meetings could be about existing VT ESL students sharing their stories in English, followed by a discussion with new VT ESL students. Guest speakers could also be invited, to provide a wider range of English exposure (e.g., accents). A mobile application could be an alternative as well, to support internationalization. That is, local students could communicate with overseas students through the application in a non-academic context. "Extracurricular" activities have been shown to be effective for second language learning [38] and these events seem to have respected the students' needs.

### 4.3. Limitations

Some participants may not have felt comfortable discussing personal concerns such as poor academic results in the focus group. For example, they may have difficulties in

reading English textbooks, so they did not perform well in courses. It is possible that these students are not willing to share such embarrassing details in the focus groups. This affects the validity of the current study. In addition, individual interviews might have affected the dynamics of the discussion and the elicitation of ideas. The results of a mixture of group and individual interviews might capture most of the students' perceptions. It is possible that most students within a group have a very similar opinion about certain aspects, and thus those with a very different opinion in the same group may not speak up to present their perspectives.

## 5. Conclusions

This study has revealed the language needs of VT ESL students and made suggestions about how proper measures can be provided to this group of students to sustain their English development from community college to university. Generally speaking, VT ESL students are not the same as ESL direct entrants. They are not highly motivated to work on their English proficiency because of many other VT issues. Their previous language training in community college prepares them well for their basic writing assignments but they need to further their writing skills for more advanced assignments (e.g., final year project). They also need to work on their speaking and listening skills for internship and other daily communication purposes. To help VT ESL students meet their needs, support measures should be short, fun, and purpose-driven. The content of support can be related to their needs, such as high-level writing skills, and daily communication for internship purposes.

**Author Contributions:** Conceptualization, D.F.; methodology, D.F., S.S.Y.C.; validation, K.C.; formal analysis, L.W.Z., G.Y.G., D.F. and S.S.Y.C.; investigation, D.F., S.S.Y.C.; resources, K.C.; data curation, K.C.; writing—original draft preparation, D.F., L.W.Z., G.Y.G.; writing—review and editing, K.C., L.W.Z., G.Y.G., D.F. and S.S.Y.C.; project administration, K.C.; funding acquisition, K.C. and D.F. All authors have read and agreed to the published version of the manuscript.

**Funding:** This research was funded by the University Grant Committee (UGC) Funding Scheme for Teaching and Learning Related Proposals (2016-19 Triennium) (PolyU6/T&L/16-19) and the Teaching Development and Language Enhancement Grant for Language Enhancement Activities (2016-19 Triennium) from The Hong Kong Polytechnic University (ELC04).

**Institutional Review Board Statement:** The study was conducted in accordance with the Declaration of Helsinki, and approved by the Ethics Committeee of The Hong Kong Polytecnic University (HSEARS20170808003 on 08 August 2017).

**Informed Consent Statement:** Informed consent was obtained from all subjects involved in the study.

**Data Availability Statement:** The data presented in this study are available on reasonable request from the corresponding author.

**Acknowledgments:** The authors are grateful to all the college transfer students for their participation in the study.

**Conflicts of Interest:** The authors declare no conflict of interest.

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
