# Peer review of "English Development Sustainability for English as Second Language College Transfer Students: A Case Study from a University in Hong Kong"

_sustainability, doi:10.3390/su141912692_

Round 1

Reviewer 1 Report

The data analysis has to be more thorough and illustrated with relevant examples. The qualitative analysis must be complemented with quantitative findings. See more specific suggestions in the attached file.

Author Response

Thank you for the comments. Please see our responses in the attached file. 

Reviewer 2 Report

I appreciate the choice of the research topic, as well as the manner of conducting research. The learning needs of VT ESL students in general, and the language needs in particular are indeed areas of little research.

The research questions are relevant and adequately answered following the study. 

The theoretical part is well developed and referenced, the cited works are varied and relevant for the hypotheses of the paper.

The article is written is fluent, coherent and correct English. I would, however, suggest minor style-related changes in order to further increase academic formality. 

Author Response

(The authors gave the same response as above.)

Reviewer 3 Report

Overall, I found this study to be well written and interesting to read. It also offers some practical advice for administrators but most likely would be more  useful to frontline teachers. I offer some advice below on how the paper could be strengthened. I am also attaching the pdf with a few notes of places where I feel some citations would be useful to backup certain claims. Hats off to the authors for a job well done. 

Introduction
Please provide some citations to backup the claims made in the first paragraph of the paper.

Please state why sustainability of English development from community college to university is crucial. 

I need more proof that the group of students you have mentioned have not received any attention from researchers. You just state that they have not been studied but is there a way for you to show lack of evidence?

Do the terms ESL students, VT ESL students, and transfer students refer to the same population or nuanced differences separate them? If they are the same, maybe you can operationalise the definition and indicate that in the paper why you use the three different terms. If they are different, that should probably be pointed out. I see for example in the paragraph starting with "Unlike VT EST students, the language needs of ESL students..."  you point out some differences. However, ESL transfer students are mentioned earlier so I think probably to make things clearer you can do something similar each time a new term is introduced. I'm not sure if further subheadings might also help readers to see the similarities or differences as well. 

No research from the same context as your study that took place?

Research Questions

Well-written and clear focuses. Claims for significance of the study are clear.

Study Design

The sentences are not well connected together. Please explain why content-analysis was selected and is suitable for analysing focus group (I guess interview data?).

Participants and Setting

How many were initially invited and what was the % resulting from those invitations? 

Data Collection

Did the same person interview all the focus groups or did more than one person do it? How would that have potentially affected the results? The same research assistant or different research assistants were present?

Could you describe the amount of data that was collected? Did you look at the number of words or something similar?

Data Analysis

Here the wording is a bit strange. It appears that Graneheim and Lundman said something about YOUR qualitative content analysis but I don't think that is your intention, it's more about the process in general. You may consider revising this sentence. 

How did the interviewer and research assistant analyse and check the coding? Could you please specify?

Were your codes all data driven or did you start out with some coding frame --maybe some concept driven route? There is no theory mentioned in the paper so I am assuming maybe it was both concept and data-driven. More details on this would be useful to those that might want to do a similar study to yours. 

Is it possible to share an example of how the coding was done or the codes to understand better where the six categories came from?

What exactly did you do during the peer debriefing and audit trial?  I know you mentioned the other manuscript but it would be nice to have some brief answers here. 

Data seems to have been analysed in Chinese but presented in English. So I suppose at some point translation took place. You may want to explain when that happened and how the integrity of the translation was ensured. 

Results

Well organised and the data used to support the findings are suitable. 

Discussion

Seems some places need references to back up some claims. I've marked these in the pdf. 

Some of the discussion section is actually implications. You may want to move the implications to another section or an easier way would be to call this section "Discussion and Implications" instead. 

Limitations

Any suggestions on how future researchers might try to overcome these limitations?

Conclusion

Good summary of the study.

Author Response

(The authors gave the same response as above.)

Reviewer 4 Report

It is recommended that the researcher not only just use one research instrument to collect and analyze the needs of VT students. With a sample size of more than 100 students, there are rigorous statistical analyses that could be run as well. Perhaps the researcher can convince the readers more with survey design as another research instrument that is quantitative.

Author Response

(The authors gave the same response as above.)

Round 2

Author Response

(The authors gave the same response as above.)
